# Anatomical and Chemical Analysis of *Moringa oleifera* Stem Tissue Grown under Controlled Conditions

**Holly M. McVea and Lisa J. Wood \***

Faculty of Environment, University of Northern British Columbia, Prince George, BC V2N4Z9, Canada
* Correspondence: lisa.wood@unbc.ca

**Abstract:** *Moringa oleifera* is a relatively well-studied ethnobotanical species, but information is limited regarding its stem anatomy and the production potential of phytochemicals from bark tissue. Knowing that variation exists in the production of chemical defenses by plants with growing conditions and with developmental stages, *M. oleifera* was grown under controlled conditions to characterize stem tissues and to determine if stem bark contained the correct phytochemical compounds to be of value in medicinal treatments. We used microscopy to characterize the stem anatomy of *M. oleifera* and analyzed stem bark extracts using FTIR and GC to identify 4-($\alpha$-L-rhamnosyloxy)-benzyl isothiocyanate (moringin) and benzylamine (moringine) in tissue. We found the stems to be in transition between juvenile and mature stages of development at 4 months old under the growth conditions used. In 7-month-old stems, we found the presence of moringin in all bark samples and did not find any moringine. These results indicate that *M. oleifera* bark of 7-month-old trees grown in greenhouse conditions may be valuable for drug development.

**Keywords:** *Moringa*; stem anatomy; bark chemistry; moringin; moringine; controlled environment

## 1. Introduction

*Moringa* species (otherwise known as miracle trees or drumstick trees) are tropical deciduous dicotyledonous trees that are distributed throughout Africa and Asia [1,2]. All parts of the plant are edible and nutritious, and they are the source of many useful compounds, making them an important famine food and source of medicinal and cleaning compounds for impoverished nations [3–5].

The leaves, seeds, and roots of *M. oleifera* are well-studied [1,6]. The leaves of *M. oleifera* are pale-green bipinnate or tripinnate and feathery with opposite ovate leaflets [1,6]. The leaves are known to have a broad array of essential nutrients in relatively large concentrations and are, therefore, a common food additive [3]. The seeds of *M. oleifera* are typically brown, roughly almond-shaped, and measure to be roughly 1.9 × 1.1 cm. The seeds are produced in large pods (the 'drumsticks') that can grow up to 50 cm in length. The seeds and pods, like the leaves, are high in nutritional quality [4]. Older roots have a vascular cambium that consists of 6–8 layers; these layers produce roundish vessel elements surrounded by xylem parenchyma cells [7]. The 3–4 layered phellogen forms rectangular or square-shaped cells, and the walls of the phellum cells are suberized. Additionally, the phelloderm is large and consists of thin-walled parenchymatous cells containing scattered groups of fibers [7]. The roots contribute to *Moringa*'s common classification as a tuber vegetable, as the roots are the most commonly eaten part of the plant [7]. Furthermore, the root bark is often harvested for various pharmaceutical and ethnobotanical uses [1,5,7].

The stem and bark of *M. oleifera* are poorly understood in relation to the other parts of the plant. To date, there has only been one anatomical analysis of mature *M. oleifera* stems [7]. There is, therefore, a gap in the literature regarding the average size and area of stem and cell tissue types across the tree's stages of growth and a further deficit of anatomical diagrams of stem sections. Each plant tissue type of *M. oleifera* has unique anatomical and phytochemical attributes, resulting in unique uses.



Originally native to the sub-Himalayan Mountains of northern India, *M. oleifera* has been cultivated for various uses in tropical and subtropical regions around the world [1]. Some uses of *M. oleifera* include biofuel production, water purification, lubrication, leather tanning, and food preparation [1,5,8]. Additionally, this species is a valuable source of phytochemicals, which assist in multiple biological activities, including oxidative DNA damage protection, promoting anti-inflammatory responses, anti-hepatoprotective processes, ulcer recovery, antibiotic immune system responses, antiperoxidative processes, and antiproliferative processes (among others) [1,5,9].

Among the many phytochemicals typically possessed by *M. oleifera* are moringine (appears chemically identical to benzylamine) and moringin (4-($\alpha$-L-rhamnosyloxy)-benzyl isothiocyanate) [5]. Moringine is an alkaloid chemical, and its presence in *M. oleifera* is the first record of a plant-produced benzylamine [10]. Moringine is considered to be a toxic compound [11], with little research elucidating the extent of its toxicity in humans. It has been used in experiments and was found to act as a potassium channel blocker, causing reduced feeding in mice [12] and resulting in a decrease in plasma-free fatty acids and water intake in rats [10]. Alternatively, moringin is a sugar derivative that has recently been shown to act medicinally against several pathologies. *M. oleifera* bark extracts containing moringin were found to effectively treat rats with aggressive breast and colorectal carcinoma [13]. Al-Asmari et al. [13] found that *M. oleifera* extracts caused a decrease in cancer cell survival and motility and an increase in malignant cell apoptosis. Moringin has also specifically been shown as promising in the treatment and prevention of ischemic stroke [14], as well as for increasing apoptosis in neuroblastoma and hepatocarcinoma cells, and as a treatment for multiple sclerosis-induced neuropathic pain [15–17].

Paikra et al. [18] reported that only the leaves of *M. oleifera* have been found to contain moringin, while seeds and roots were found to contain moringine. To date, we could not find any specific literature describing the phytochemistry of bark that included both testing for moringin (potentially beneficial) and moringine (potentially harmful).

In natural environments, quantities of secondary metabolites such as moringin and moringine vary with environmental conditions; changes in solar radiation, temperature, nutrient availability, water availability, and biotic competition may all influence chemistry [5,19,20]. In optimal environments, plants tend to invest more energy into growth and reproduction than into protective anti-herbivory measures (such as toxic secondary metabolites) [5,19]. Trees growing in optimal environments may produce fewer toxic compounds, such as moringine, as there is little need for anti-herbivory measures. No information currently exists on how levels of moringin or moringine in *M. oleifera* may be altered by the environment.

To add to the current literature, we sought to depict the distribution and size of various tissue types within *M. oleifera* stems grown in a greenhouse under specific environmental conditions (Objective 1). We also sought to evaluate the efficacy of the bark as a source of secondary plant compounds when grown under controlled conditions, considering specifically the presence of moringin and moringine (Objective 2).

## 2. Methods

### 2.1. Growing Conditions

All *M. oleifera* trees used for this study were grown at the I.K. Barber Enhanced Forestry Lab (EFL), University of Northern British Columbia (UNBC). For the *M. oleifera* trees used in anatomical analyses, germination methods were derived from *Moringa* Farms [21]. Seeds of Indian origin were imbibed in a zip-lock bag (left slightly unsealed to allow for airflow) filled with water for 24 h. After the imbibing period, the seeds were removed from the water, dried on paper towel, and placed in a closed paper bag left in a cupboard above a stove to provide a dark warm germination environment. After 14 days, most of the seeds had germinated, and the seeds were planted 2 cm deep into soil. The soil mixture consisted of 107 L peat, 20 L perlite, 20 L vermiculite, 1/3 c coir soil enhancer, 1/3 c dolomitic lime, 1/3 c MicroMax nutrients, and 2/3 L slow-release nutrients (14-14-14). The trees were

watered to saturation approximately twice weekly with plain water for the first month and with water containing 14-14-14 fertilizer at every watering thereafter, as signs of nutrient deficiency, including chlorosis, were noticed after 1 month of growth. During watering, the trees were monitored for signs of diseases to ensure none were present. The trees were grown in a growth chamber at 30 °C during the day (0600 to 2200 h) under LED grow lights and cooled to 28 °C between 2200 h and 0600 h with the lights turned off. After 4 months of growth, the trees were cut down and immediately underwent anatomical analysis.

The methods used for growing the trees utilized in chemical analyses were developed by Morgan et al. [8]. Seeds were planted 2.54 cm deep into a soil mixture prepared by mixing 20 L coir, 20 L coarse sand, and 20 L peat with 60 g of slow-release nutrients (14–14–14); there was also an addition of 3 tablespoons of dolomite to the soil prior to planting. High-pressure sodium (HPS) supplemental lighting was supplied to the trees each day between 0600 to 2200 h. The trees were kept at 24 °C within the housing greenhouse bay and maintained at a relative humidity ranging from 20 to 40%. During the night (2200–0600 h), the lights were turned off and the bay was cooled to 18 °C. The trees were watered approximately three times per week to the point of saturation. These procedures were followed for 7 months, at which point the trees were cut down and frozen until the bark could be peeled from the stems. The bark was then kept frozen until the time of phytochemical analysis.

### 2.2. Anatomical Analysis

Cross-sections were obtained from five freshly harvested 4-month-old *M. oleifera* plants by hand using a razor blade. The cross-sections were stained with toluidine blue (TBO) and then photographed and examined using a DS-Ri2 Eclipse FN1 Nikon light microscope under $40\times$ magnification. The cross-section images were analyzed using NIS-Elements Basic Research (v.5.10.01 64-bit) software. Tangential width measurements (at the widest point, from the outside of the cell wall on one side to the other) were recorded for dilated phloem rays, vessel elements, and xylem rays. Radial width measurements were recorded for the periderm, true phloem tissue, and vascular cambium (at the widest point, from the outside of the cell wall on one side to the other). Lastly, small, localized pockets of sclerenchyma tissue within the cortex were measured by tracing the perimeter of the cell walls; the imaging software then algorithmically calculated the estimated area within the tracing. Raw data measurements were collected and loaded into Microsoft Excel.

### 2.3. Chemical Analysis

The extraction methods used were derived from Oluduro et al. [22]. The bark was removed from 19 frozen samples of 7-month-old *Moringa*. Once the bark had been removed, it was ground using liquid nitrogen and an IKA A11 basic analytical mill. The mill was cleaned between samples to avoid cross-contamination. Once the bark was ground, ethanolic extracts were prepared for each of the 19 samples. A total of 5 g ($\pm 0.10$ g) of plant tissue (with an average moisture content of 39.7 % (SE = 1.39 %, CV = 0.111) were mixed with 100 mL of 90% ethanol. The solutions were then left at room temperature and shaken for 30 min a day at 120 rpm for 3 days and then left undisturbed at room temperature for an additional 2 days. Once the extraction time had fully elapsed, the suspension solutions underwent gravity filtration through #1 Whatman paper until clear (at least two filtrations were necessary). The extractions were then rotovapped at room temperature to remove some of the excess ethanol and water. The concentrated extracts were then filtered through 0.45 µm microfilters via syringe. The finished extracts were then stored in air-tight 2 mL vials at 4 °C.

Extracts were analyzed using Fourier-transform infrared spectroscopy (FTIR) on a Bruker ALPHA II with a platinum ATR module to detect if moringin or moringine were present. Determining the presence of the two compounds consisted of comparisons to FTIR of standard-grade moringin (molecular weight = 311.35 g/mol) and moringine (molecular weight = 107.15 g/mol). The moringin standard was >98.0% pure, obtained from ChemFaces (CAS No. 73255-40-0, Catalog No. CFN89445), and the moringine

standard, in the form of benzylamine >99.0% pure, was obtained from Sigma-Aldrich (CAS No. 100-46-9, Catalog No. 13180). Once the qualitative phytochemical presence of the two compounds of interest were determined, the extracts were quantitatively analyzed using gas chromatography (GC) on an Agilent 6890GC at Northern Analytical Laboratory Services (NALS) at the University of Northern British Columbia's Prince George campus. The 19 samples were run with a ValcoBond VB5, 30 m × 0.25 mm × 0.25 μm film thickness column, with a constant flow rate of 1.7 mL/min using helium carrier gas at a 20:1 split ratio, with an oven programmed at 120 °C for 1 min, ramped 10 °C/min to 250 °C then ramped 25 °C/min to 300 °C and then held for 4 min at 300 °C.

*2.4. Data Analysis*

Descriptive statistics were calculated for the anatomical measurements across the 4-month-old trees using Microsoft Excel 2020 and IBM SPSS version 26 statistical analytics software. When possible, all 5 samples were used in the calculation; unfortunately, in some of the samples, the phloem rays were interconnected and/or the sclerenchyma tissues ran the circumference of the stem and were not possible to measure.

A distribution report was produced to summarize the presence–absence data resulting from the FTIR analysis; sample peaks were matched to the standard outputs and to the output of pure ethanol to determine if either moringine or moringin were present in the ethanolic extracts. Using GC, compound determinations were based on retention times with standards. The peaks of the moringin standard were integrated over the whole range of peaks to acquire concentration data; standard concentration data were plotted against area to determine correlation and for inaccuracy correction ($R^2$ = 0.992).

Since this was discovery-based research and not a true experiment, we did not make any statistical comparisons aside from obtaining average measurements of anatomy and chemical concentrations.

## 3. Results

*3.1. Moringa Stem Anatomy*

At the time of harvest (4 months old), the trees were transitioning from the juvenile stage to maturity; Nielsen [23] reported that it can take up to 8 months for the tree to fully mature. This juvenile–mature transition was evident by the presence of sclerenchyma tissues, which were visible as separate regions of tissue above and beside thick pockets of phloem (indicating some immaturity), combined with relatively thick xylem tissues connected in a complete circle within the circumference of the stem (indicating a level of maturity) (Figures 1 and 2). The transitioning trees had an average of 83 xylem rays/stem, with the number of rays ranging from 32 to 104 (SD = 29.04 rays, CV = 0.3490). Each of the tissue and cell types measured in the transitioning stems can be observed in Figures 1 and 2. In addition to the diameters, the sclerenchyma tissue pockets had an average area of 296,648 μm$^2$ but ranged from 23,051 μm$^2$ to 889,283 μm$^2$ (SD = 192,290.0, CV = 834.2). The radial phloem and tangential dilated phloem ray measurements had the greatest range (SD = 237.29 μm and 112.83 μm). Individual comparisons between tissue and cell measurements can be observed in Figures 3 and 4. These measurements highlight the consistency across individuals in periderm and vascular cambial radial diameters and the variability in the radial diameter of phloem tissue (Figure 3). Figure 4 illustrates that the vessel elements are relatively consistent in tangential diameter across individuals but show a range of sizes within any given individual. Figure 4 also shows the relative consistency in xylem ray tangential diameters across and within samples.

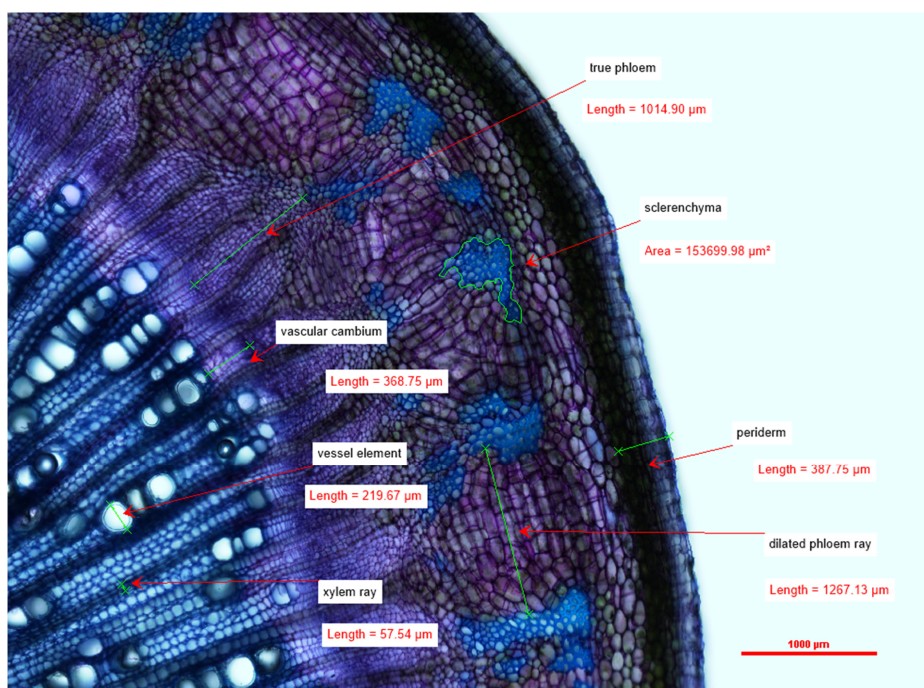

**Figure 1.** An example of a quarter-transverse section of a *Moringa oleifera* stem, stained with TBO, transitioning from the juvenile developmental phase to maturity observed under 40× magnification depicting prominent tissue and cell types and their measurements (measured at the widest point, from the outside of the cell wall on one side to the other).

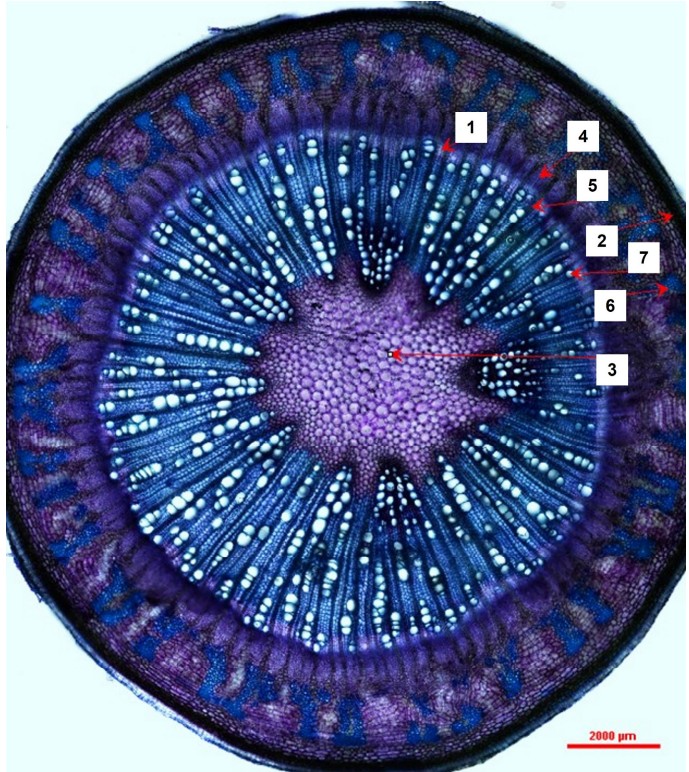

**Figure 2.** A transverse section of a *Moringa oleifera* stem transitioning from the juvenile developmental phase to maturity observed under 40× magnification depicting measured prominent tissue and cell types (1 = vascular cambium, 2 = periderm, 3 = pith, 4 = true phloem, 5 = xylem ray, 6 = sclerenchyma, 7 = vessel element).

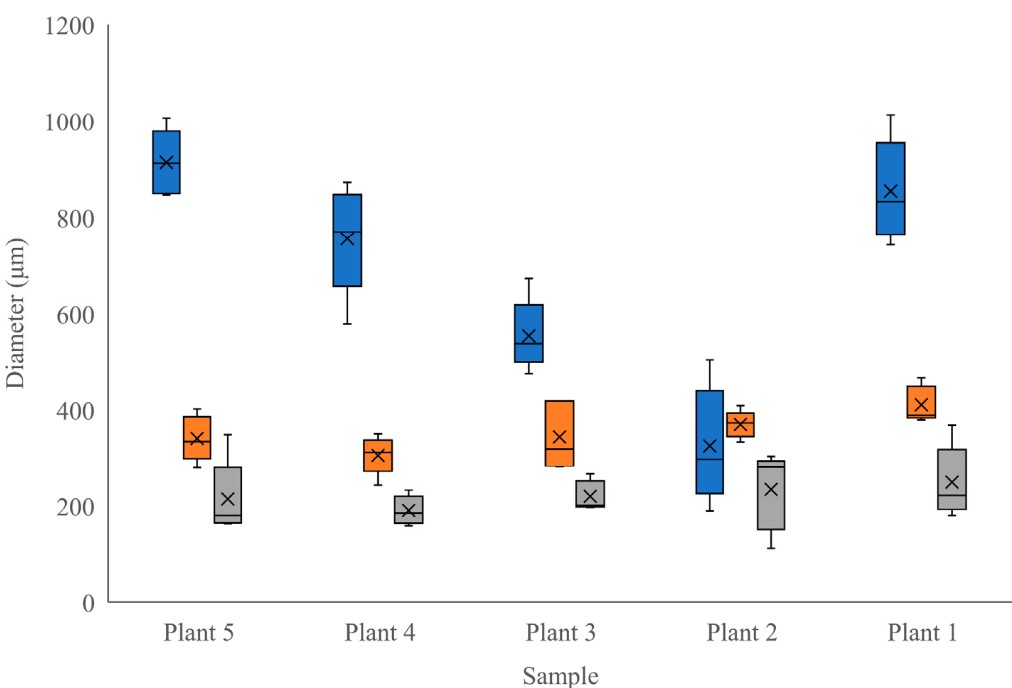

**Figure 3.** A comparison of the distributions of radially-measured tissues diameters in 5 stems of 4-month-old *Moringa oleifera* (measured at the widest point, from the outside of the cell wall on one side to the other) between tree samples (blue = true phloem, orange = periderm, grey = vascular cambium).

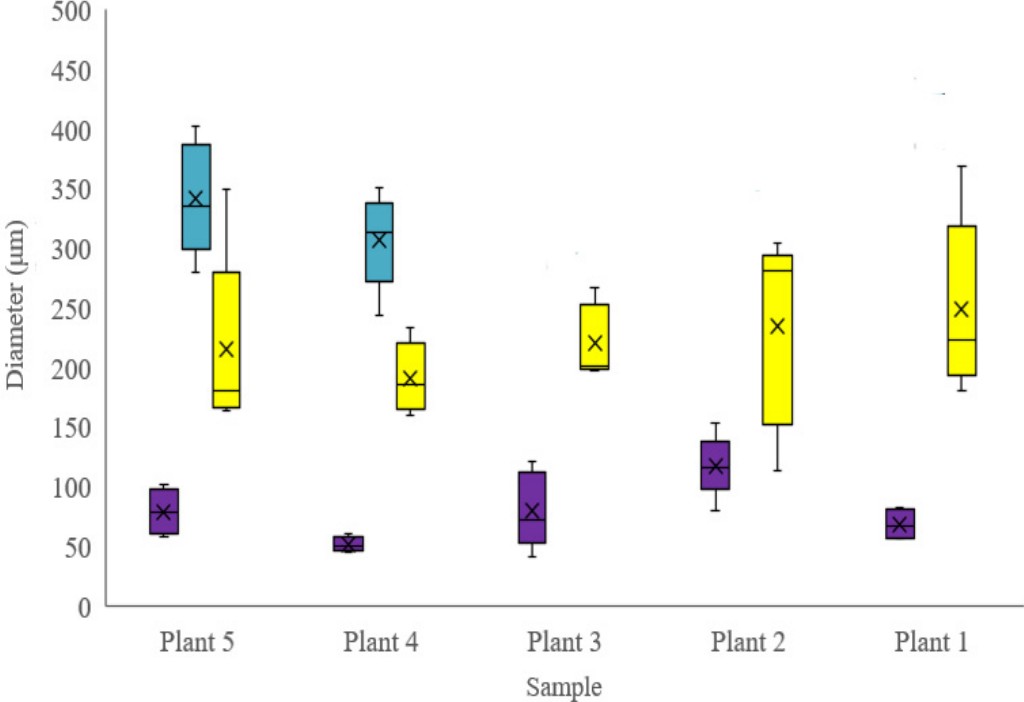

**Figure 4.** A comparison of the distributions of tangentially-measured tissues and cells diameters in 5 stems of 4-month-old *Moringa oleifera* (measured at the widest point, from the outside of the cell wall on one side to the other) between tree samples (cyan = dilated phloem ray, yellow = vessel element, purple = xylem ray).

### 3.2. Chemical Findings

The FTIR analyses indicated that all 19 samples of stem bark from 7-month-old *Moringa* were nearly identical in their qualitative chemical composition. The FTIR indicated the presence of moringin in all 19 samples, indicated by the matching peaks in the samples and the standard (Figure 5). The peaks of interest can be observed at between 850–1500 cm$^{-1}$ and at 1920 cm$^{-1}$ (Figure 5); the remaining matching peaks are a result of the –OH group in the ethanol matching with the –OH in the moringin between 2800 cm$^{-1}$ and between 3400 cm$^{-1}$. The FTIR outputs further demonstrated the lack of moringine in all 19 samples; the lack of matching peaks can be observed in Figure 6.

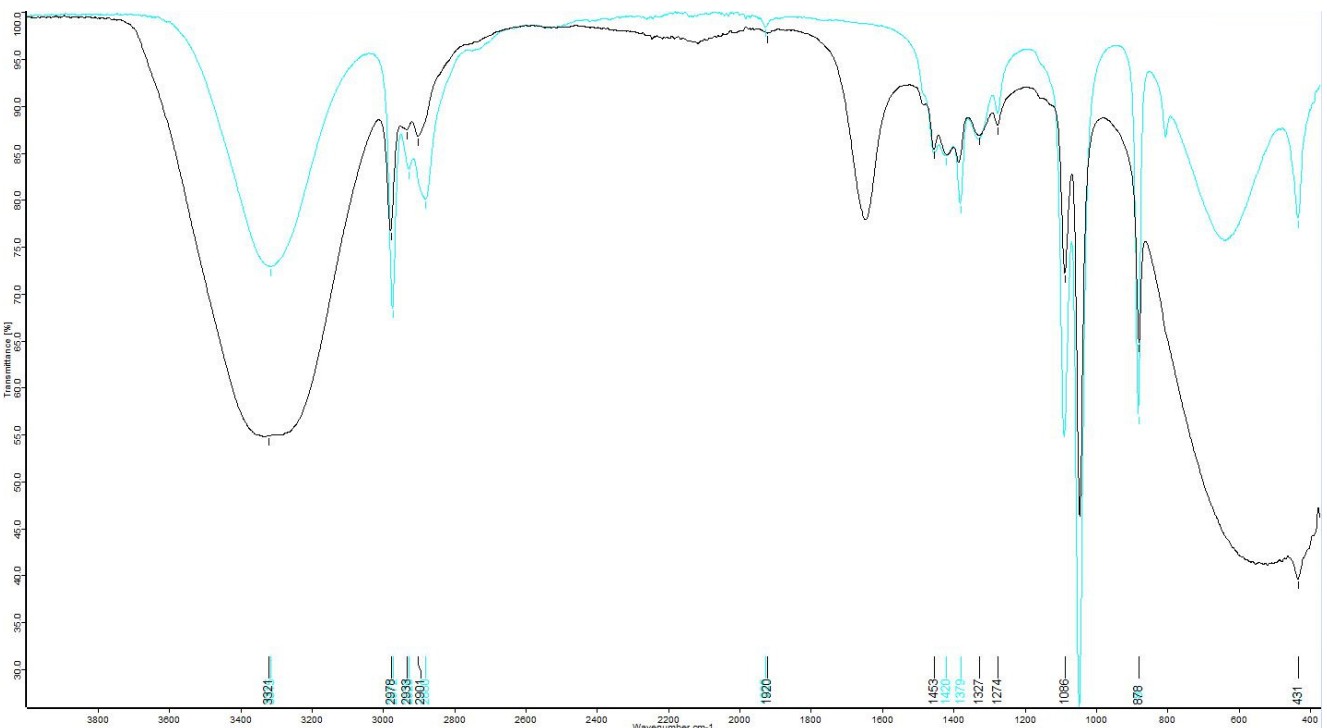

**Figure 5.** FTIR output, measuring percent transmittance over a range of wavelengths (cm$^{-1}$), comparing a representative ethanolic bark extract of *Moringa oleifera* (black) to standard-grade moringin (cyan). The many commonalities in peaks between the representative extract and the standard suggest that there is moringin present in the extracts.

The GC analyses conducted were able to quantify the amount of moringin present in the ethanolic extracts and confirm the lack of moringine (Figure 7). In ranges of peaks of interest, the chromatograms were well resolved; the moringin standard produced a cluster of peaks between 15.2 and 15.4 min. The chromatograms showed that the samples contained multiple compounds, which was expected, and is typical of plant material. A large peak was evident for the solvent (ethanol), and smaller peaks were detected at various points, including at the appropriate retention time for moringin (aligned with the standard). The moringine standard produced a large, narrow peak at 2.145 min, indicating its volatility. Samples did not show peaks corresponding to the moringine standard (Figure 7). The GC indicated that there was an average moringin concentration of 80.39 ug/mL in the extracts (SE = 2.06 ug/mL) with a range in concentrations between 8.29 ug/mL and 151.16 ug/mL (Table 1).

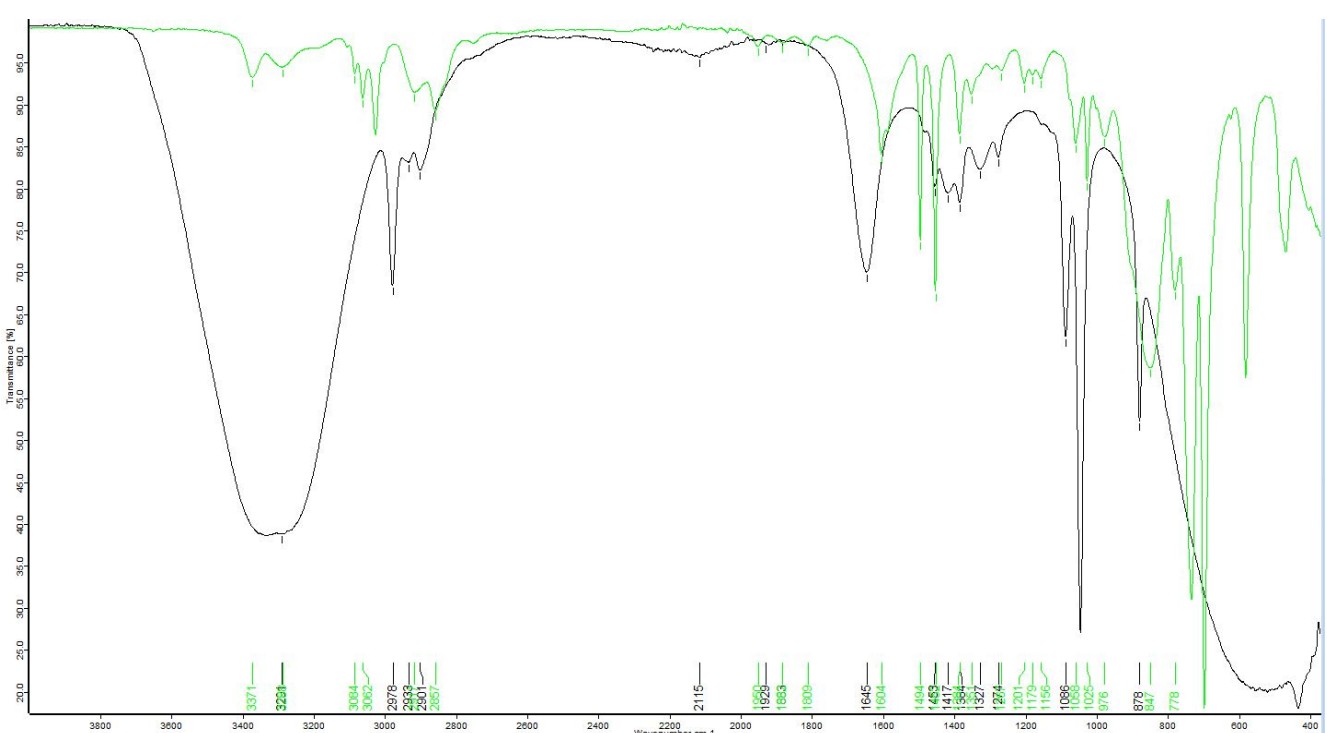

**Figure 6.** FTIR output, measuring percent transmittance over a range of wavelengths (cm$^{-1}$), comparing a representative ethanolic bark extract of *Moringa oleifera* (black) to standard-grade moringine (green). The minimal commonalities in peaks between the representative extract and the standard suggest that there is no moringine present in the extracts.

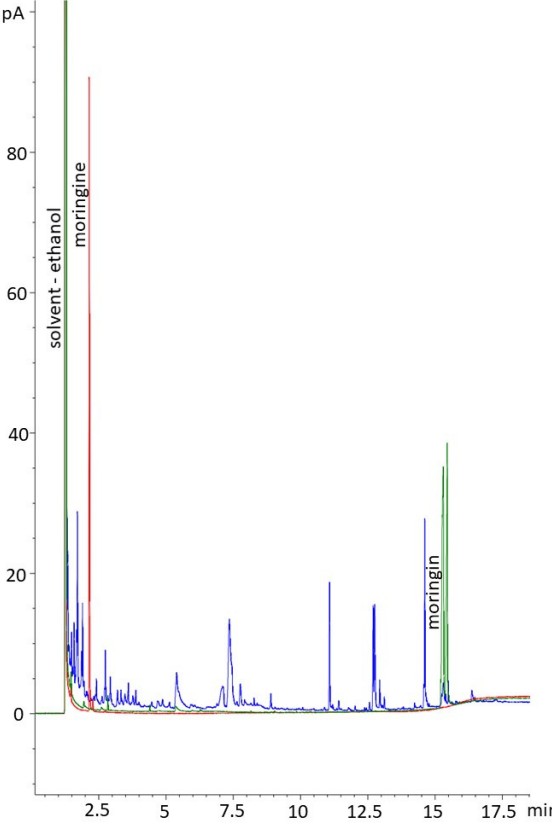

**Figure 7.** Chromatogram showing the moringin standard (green), moringine standard (red), and a representative *Moringa oleifera* sample extract, extracted from the *M. oleifera* bark (blue).

**Table 1.** Concentrations of moringine and moringin calculated using gas chromatography (forced zero) and the peak area for each of the 19 bark samples tested of *Moringa oleifera*.

| Sample ID | Benzylamine Area | Moringin Area | Benzylamine (ug/mL) | Moringin Conc (ug/mL) |
|---|---|---|---|---|
| Y18 | 0.000 | 6.814 | 0.000 | 46.693 |
| Y18-2 | 0.000 | 7.946 | 0.000 | 54.450 |
| Y63 | 0.000 | 16.786 | 0.000 | 115.031 |
| Y56 | 0.000 | 18.340 | 0.000 | 125.679 |
| Y61 | 0.000 | 6.901 | 0.000 | 47.290 |
| Y41 | 0.000 | 7.242 | 0.000 | 49.625 |
| Y29 | 0.000 | 14.365 | 0.000 | 98.440 |
| Y49 | 0.000 | 17.167 | 0.000 | 117.641 |
| Y24 | 0.000 | 5.274 | 0.000 | 36.143 |
| Y30 | 0.000 | 6.051 | 0.000 | 41.465 |
| Y48 | 0.000 | 17.477 | 0.000 | 119.764 |
| Y38 | 0.000 | 10.742 | 0.000 | 73.610 |
| Y04 | 0.000 | 16.662 | 0.000 | 114.180 |
| Y13 | 0.000 | 7.377 | 0.000 | 50.550 |
| Y07 | 0.000 | 22.058 | 0.000 | 151.162 |
| Y65 | 0.000 | 16.447 | 0.000 | 112.710 |
| Y70 | 0.000 | 10.085 | 0.000 | 69.113 |
| Y52 | 0.000 | 13.946 | 0.000 | 95.567 |
| Y15 | 0.000 | 1.210 | 0.000 | 8.292 |

## 4. Discussion

We sought to depict the distribution and size of various tissue types within *M. oleifera* stems grown in a greenhouse under specific environmental conditions (Objective 1). In a study by Vyas [7], a brief overview of *Moringa oliefera*'s stem anatomy was previously documented. They described young stems as having 16–18 vascular bundles, a large parenchymous pith, and a pericycle composed of alternate groups of parenchyma cells and fibers; these groups eventually form a circular band in mature stems [7]. Vyas [7] further describes mature stems' vascular cambia (VC) that produce large amounts of secondary xylem, which consists of uniseriate xylem rays, roundish vessel elements, and lignified thick-walled fibers; the VC also produces small amounts of secondary phloem. We add to this description by providing measurements for a variety of tissue and cell types in juvenile–mature transitional tissues of *Moringa oliefera*. This growth stage highlights the development of the ring of secondary tissues, including sclerenchyma. Interestingly, the radial diameter of the true phloem showed much more variability between individuals than did other tissue types. Phloem is the main transport tissue for photosynthates, metabolites, and other compounds and generally displays a high level of plasticity in response to environmental factors [20]. Given that these trees were all grown in the same, highly controlled environment, it is unlikely that atmospheric conditions were the reason for this variability. Seeds were also all from the same source, although they could have been derived from different parent trees. Since our plants did suffer from a nutrient deficiency at one point during their growth period, it is possible that the severity of this stress was different among individuals and, therefore, created a difference in phloem development [20], contributing to the variability shown among phloem tissues between our samples (Figure 3). The higher degree of consistency in the other tissue and cell sizes between individuals indicates that tissues other than phloem were relatively unaffected by

the period of mineral deficiency (Figures 3 and 4). Our description of the stem anatomy of juvenile–mature transitioning *M. oleifera* grown under controlled conditions (Objective 1) adds to the currently available literature describing the anatomy of this species.

We qualitatively and quantitatively confirmed that 100% of the samples contained moringin using FTIR and GC, indicating that moringin is consistently produced in *Moringa oleifera* bark in trees grown under controlled greenhouse conditions. Thus far, moringin has largely been extracted from the seeds of *Moringa* for medicinal use [14,24,25]. The bark extracts we produced contained an average of 0.08 mg/mL and up to 0.15 mg/mL of moringin, but the samples consisted of an average of 39.7% water; therefore, the concentration of moringin in solution was diluted. Further refinement of the extraction process from bark could likely yield concentrations that are useful in medicine, for example, for the treatment of spinal cord injury and ischemic stroke [14,25]. Moringin was administered to rats at 3.5 mg/mL daily, producing neuroprotective properties and reducing oxidative stress and inflammation [14].

The double confirmation (through both FTIR and GC) that none of the bark samples contained moringine indicates that a greenhouse environment did not encourage the production of the potentially toxic compound [5,10–12]. In outdoor conditions, there are greater temperature, moisture, and nutrient fluctuations than under greenhouse conditions. Growing under optimal conditions and not being exposed to the dangers of herbivory, plants are less likely to produce defensive or stress-related secondary metabolites. Furthermore, plant chemical defenses vary greatly with age, and there is a particular reduction in chemical defenses during the transitional phase between being a juvenile and maturity [26]. Given that it can take up to 8 months for *M. oleifera* to mature [23], our research supports that growing *M. oleifera* in stable greenhouses for 4–7 months yields plants that do not have a risk of containing moringine. This is an important conclusion for those looking to produce *Moringa* plants for the treatment of pathologies.

We sought to evaluate the efficacy of the bark as a source of secondary plant compounds when grown under controlled conditions, considering specifically the presence of moringin and moringine (Objective 2). This research suggests that by producing fast-growing *M. oleifera* in optimal greenhouse conditions, trees can produce bark within 4–7 months, from which the extract is of good quality for use in medicinal treatments. This production can be conducted anywhere in the world, as illustrated by our location in northern Canada, and is not limited to tropical growing conditions, as would be the case in outdoor cultivation.

## 5. Conclusions

*M. oleifera* trees at 4–7 months of age contain transitional juvenile–mature anatomy. These tissues show the development of secondary tissues and relatively large and variable amounts of phloem, possibly reflective of the nutrient regime. *M. oleifera* grown under optimal growing conditions produced no moringine, but they maintained the production of moringin, likely due to the constantly available resources and lack of competition and herbivory, thus making phytotoxins superfluous. Future improvements could be made to this research through the use of another standard for GC to possibly produce a cleaner peak and through the analysis of a more concentrated extract. Overall outcomes of this study contribute to the effort to understand this 'miracle tree' in its entirety and to elucidate bark as an alternative to seed use for the production of moringin. Future studies could add to our findings by narrowing the growing conditions of *M. oleifera* to maximize moringin yields, optimizing the extraction process from bark, and using trees of different ages to determine if and when moringine is produced under alternative conditions. Furthermore, comparing trees grown in outdoor conditions to controlled conditions in a true experiment would add to current knowledge.

**Author Contributions:** H.M.M. conducted the literature review, carried out all methods, analyzed results, and wrote the initial manuscript draft as an undergraduate thesis project. L.J.W. established the methods for use, provided materials and experimental space, supervised the research, added to

the literature review, and revised the writing for manuscript submission. All authors have read and agreed to the published version of the manuscript.

**Funding:** This research received no external funding. The APC was funded by the University of Northern British Columbia.

**Institutional Review Board Statement:** Not applicable.

**Informed Consent Statement:** Not applicable.

**Data Availability Statement:** Data are kept at the University of Northern British Columbia, and are available upon request. Please email the corresponding author.

**Acknowledgments:** This research received no funding; however, we would like to acknowledge Chris Opio for his expertise and guidance in the research and for providing in-kind contributions of the 7-month-old stem samples and seeds. We would also like to acknowledge Jasneek Manhas and Katie Tribe for their help in the laboratory. Lastly, we would like to thank the UNBC Chemistry Department (Beth Gentleman, Kaila Fadock, and Todd Whitcombe) and NALS (Charles Bradshaw) for the contributions of their laboratories, equipment and supplies, and expertise.

**Conflicts of Interest:** Authors declare no conflict of interest.

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
