# Peer review of "Anatomical and Chemical Analysis of Moringa oleifera Stem Tissue Grown under Controlled Conditions"

_horticulturae, doi:10.3390/horticulturae9020213_

Round 1
Reviewer 1 Report
The manuscript entitled "Anatomical and chemical analysis of Moringa oleifera stem tissue grown under controlled conditions" presents interesting results about the production of moringin (potentially beneficial) and moringine (potentially harmful) in M. oleifera bark. However, the manuscript does not follow the journal format. The introduction does not need to be divided into several sections and there is no need for an extra section showing the objectives of the work. As it is, the manuscript seems more like a thesis than an article.
Other proposed changes:
- Growing conditions - The text contained repeated sentences. Please, check again this section.
Figure 3 - the colors reported in the legend do not correspond to the colors used to prepare the figure. Please, clarify this aspect.
Results - Although in the conclusion section the authors have stated "Future improvements could be made to this research through use of an another standard for GC to possibly produce a cleaner peak, and through the analysis of a more concentrated extract.", in my opinion, the authors should provide a representative GC chromatogram of the stems analyzed as well as the moringin and moringine chromatograms. A table showing the quantification of moringin in the 19 analysed samples should also be provided. Moreover, what is the explanation for the wide range of moringin concentration obtained among samples? Please, give some clues about this subject.
For those reasons, I recommend major revision to this manuscript.
Author Response
We thank you for your thorough review. We have adjusted the introduction to reflect the concern. Subtitles have been removed and the content has been condensed to reflect a more appropriate journal format.
The section in methods on growing conditions has been edited to remove any unnecessary repetition.
Figure 3 caption lists colours of the series as blue, red, and green, which are accurate to the figure, so we are unsure of what the reviewer means here.
We have added a table showing the quantification of moringin in each of the 19 samples as requested (Table 1). We have also provided a representative chromatograph from the GC output which overlays the standard peaks onto a representative sample we tested (Figure 7).
Reviewer 2 Report
The manuscript in reference describes anatomical and FTIR/GC-based chemical characterization of Moringa oleifera grown under controlled conditions. However, it has several shortcomings that limit further consideration.
1. In general, the main criticism is related to the fact that this study comprises information that can not be considered enough to meet the publishing requirements of the journal. In other words, it provides low-quality information to the researchers and readers.
2. The logic and neatness of the introduction need to be further improved. It is too long and provides unnecessary information about the study's aim and scope.
3. The aim and scope of the study are unclear. Although a section is dedicated to objectives, the aim and scope are still unclear.
4. Materials and methods are difficult to read and need to be detailed. In addition, the experimental design is not correctly described.
5. The chemical information is incorrect since the chemical analysis is wrongly afforded. There are other better analytical platforms to evaluate the chemical composition and metabolite presence. FTIR is very general, and GC can be challenging to detect minor glucosinolates from M. oleifera (as in the case of the undetected moringin by FTIR) and worst if they come from ethanolic extracts. Information on quality, purity, and brand of chemical standards are missing. In addition, findings from FTIR are exploited (although they are not confirmatory), and GC-derived results are poorly used and explained.
6. The discussion section is not well-oriented, and this is a result of an incorrect aim and scope.
7. The conclusion section is too long, and several passages summarize the results. Better conclusions about conceptual findings from the mechanistic point of view are missing.
Author Response
We thank you for your thorough review. We have adjusted the introduction to reflect the concern. Subtitles have been removed and the content has been condensed to reflect a more appropriate journal format.
Through the refinement of the introduction and the omission of some extraneous information we hope that the aim and scope have become clear. We have outlined our aims in lines 93-97.
We have edited the methods section to clarify our process. We have included the detailed information on the chemical standards used and models of equipment (lines 225-240).
We followed standard protocols from other publications for extraction procedures, as stated in the methods, and therefore we do not feel that these methods are inappropriate. FTIR was used to detect presence/absence of the two chemicals investigated, and it is an appropriate method to use for this purpose. Because FTIR gives a general idea of components in complex mixtures we used GC to quantify. GC is a well-known method for the quantification of compounds like those we sought to measure. We added more information about the GC output, including Figure 7 and Table 1 to better explain the results of the GC.
We have directly referenced our aims in the discussion to ensure that our point is clear.
We have revised the conclusion to reduce repetition of results and to include more mechanistic points of view as suggested.
Reviewer 3 Report
The paper entitled "Anatomical and chemical analysis of Moringa oleifera stem tissue grown under controlled conditions" is very interensting and carefully detailed. This work fills a gap in the literature on the topic, and it has been performed properly.
I have only few brief suggestions for the authors:
- please revise the formatting of figures' captions, according to the guidelines provided by the Journal.
- Even if the sections that can be used in the manuscript, reported in the template, can be eventually changed, I suggest not to divide the introduction into paragraphs, and not to separate the aim of the work from the introduction. Moreover, the paragraph "Anatomy of Moringa oleifera" is perhaps beyond the aim of the work, and hence it could be reduced.
- please provide more detailed information on Excel and SPSS statistical analytics softwares versions used in the work, as well as the model of FTIR and GC.
Author Response
We thank you for your thorough review. We have edited the introduction to remove subheadings and to condense the text. We provided the versions of the statistical software used. We have included the model information for the FTIR and the GC and the details of the standards used on lines 225-240.
Reviewer 4 Report
The manuscript "Anatomical and chemical analysis of Moringa oleifera stem tissue grown under controlled conditions" by the authors Holly McVea and Lisa June Wood is devoted to the study of the structural features of the stem bark of the Moringa oleifera plant, which is important for traditional medicine and obtaining dietary supplements. The work contains few methodological approaches, and has many significant shortcomings. In its present form, the work cannot be accepted, since a lot of corrections will be required and there are methodological errors that are unlikely to be corrected quickly.
Let's start with technical issues. 1) The manuscript is not designed according to the rules of the MDPI journal. Namely: the parts of the manuscript must be numbered; parts of the manuscript cannot differ from the established rules (authors enter, for example, "Objectives"; subchapters must also be numbered; references must be placed in square brackets and contain numbers corresponding to the list of references.
2) The authors violate the traditional rules for writing articles, in part of the manuscript text they write the pronouns "I", in the part "We" (such forms should be used only in extreme cases). 3) The language of the article cannot be recognized as scientific, in addition to errors, there are unused "terms" and inadmissible definitions, for example "individual", the English grammar also needs to be radically corrected.
However, it is not these essential remarks that force us to reject the publication or offer to resubmit it after correction. Most significantly, the authors do not explain in any way why they had the right to avoid the use of at least some control. Even if this is a rare plant, this cannot be an excuse, because it was possible to study the dynamics of changes in individual tissues or experiment with various conditions, for example, use paraquat or at least hydrogen peroxide, or inflict mechanical damage ... But there is nothing like that. Against this background, the authors conclude that the differences (what is compared with what?) are associated with the absence of damage by animals (in this case, no data are provided on the control of damage, for example, by aphids or leaf-eating, which could cause a similar effect). In addition, the reason for such frequent watering is not clear to me. For this drought-resistant plant, such watering is clearly not the norm, that if the results shown by the authors are caused by excessive watering.
In general, we have to state that this work was performed and formatted with significant errors, and cannot be accepted for publication.
Author Response
We thank you for your thorough review. We have corrected many issues with the language and flow of the manuscript throughout the body of the document. We believe that the English language concerns have been rectified. Parts of the manuscript have been numbered and references have been reformatted.
Although traditionally articles are written in third person, in the last 10-20 years it has become perfectly acceptable, if not encouraged to write in first person and in active voice. The authors firmly believe that writing in third person is no longer the standard of scientific writing and that exclusive third person conventionality is outdated.
We do not have a control because we did not conduct an experiment. This was discovery-based research to describe a species, which is a necessary step prior to the initiation of experimentation (and we do make recommendations for next steps in our conclusion, which would include comparative experimentation). In order to make this clear, we have revised our methods, results and discussion sections so that our approach is obvious, and we have avoided any language that could be mis-represented as comparisons in an experiment.
Our results are not related to excessive watering as the reviewer may indicate. The plants were grown in pots, and soil dried quickly requiring frequent watering. Watering was done to a necessary schedule to keep plants at their optimal growth.
Round 2
Reviewer 1 Report
Dear authors,
Thank you very much for taking into account my comments. Regarding Figure 3, in my computer I see the image with blue, orange and grey colors, but probably colors have been changed in the pdf file and the original figure is blue, red and green. Please, just clarify this because I do not see red and green but orange and grey.
In Figure 7, please change "chromatograph" by "chromatogram".
Author Response
Thanks again. Yes, we see now that in the edited and set version of the manuscript the colours in Figure 3 were altered. We have adjusted the legend colours so they now coordinate with what is shown. In Figure 7, we have made the change to “chromatogram”.
Reviewer 2 Report
The authors adequately addressed my comments, so the manuscript improved and can be followed properly. I have only two minor comments about this revised manuscript version:
1. Figures 1 to 7 are disproportionately large, which looks problematic for the journal's format. Therefore, they could be adjusted to a better size:aspect ratio so that the information and text are adequately presented.
2. In table 1, the metabolite concentration can be expressed in ppm or micrograms per milliliter (ug/mL) to provide more comprehensive data.
Author Response
Thanks again for your feedback. We have replaced Table 1 values with ug/mL as suggested.
I am not sure if the journal editor will adjust the figure sizes, or if that should be done on my end. I will connect with the editor about this.
Reviewer 4 Report
Manuscript Anatomical and chemical analysis of Moringa oleifera stem tissue grown under controlled conditions by
authors/ Holly McVea, Lisa June Wood has been redesigned and can be accepted.
However, there are still some questions. For example:
in figure 2, the number three is surrounded by squares
Figure 3, 4 and table 1 do not have p values in the caption to understand the statistical limitations.
Questions about the use of language and pronouns can be left to the editors.
When these remarks are corrected, the manuscript may be printed.
Author Response
Thank you again for your comments. The number 3 in Figure 2 has been edited to remove the squares.
We have not included p-values in Figures 3, 4 and Table 1 because we did not conduct any statistical comparisons. We are simply commenting on variability in our discussion of Figures 3 and 4, which stems from the visual trends shown in the data, and Table 1 is raw data, which was not statistically analyzed apart from obtaining averages.